# Anti-dsDNA B-Cell ELISpot as a Monitoring and Flare Prediction Tool in SLE Patients

**DOI:** 10.3390/jcm12041295

**Published:** 2023-02-06

**Authors:** Albert Pérez-Isidro, Marc Xipell, Arturo Llobell, Noemí De Moner, Gema M. Lledó, Ricard Cervera, Sergio Prieto-González, Luis F. Quintana, Gerard Espinosa, Mila García-Ormaechea, Estíbaliz Ruiz-Ortiz, Odette Viñas

**Affiliations:** 1Department of Immunology, Centre de Diagnòstic Biomèdic (CDB), Hospital Clínic, 08036 Barcelona, Spain; 2Institut d’Investigacions Biomèdiques August Pi i Sunyer (IDIBAPS), 08036 Barcelona, Spain; 3Department of Nephrology and Renal Transplantation, Hospital Clínic, University of Barcelona, 08036 Barcelona, Spain; 4Reference Centre for Complex Glomerular Disease (CSUR) of the Spanish Health System, 08036 Barcelona, Spain; 5Department of Autoimmune Diseases, Hospital Clínic, University of Barcelona, 08036 Barcelona, Spain; 6Reference Centre for Systemic Autoimmune Diseases (CSUR) of the Spanish Health System, 08036 Barcelona, Spain; 7Lime Tree Surgery NHS, Worthing BN14 0DL, UK

**Keywords:** systemic lupus erythematosus, SLE biomarkers, SLE nephritis, SLE disease activity, antibody secreting cells ELISpot, anti-dsDNA autoantibodies

## Abstract

Anti-dsDNA autoantibodies quantification and complement levels are widely used to monitor disease activity in systemic lupus erythematosus (SLE). However, better biomarkers are still needed. We hypothesised whether the dsDNA antibody-secreting B-cells could be a complementary biomarker in disease activity and prognosis of SLE patients. Fifty-two SLE patients were enrolled and followed for up to 12 months. Additionally, 39 controls were included. An activity cut-off (comparing active and non-active patients according to clinical SLEDAI-2K) was established for SLE-ELISpot, chemiluminescence and *Crithidia luciliae* indirect immunofluorescence tests (≥11.24, ≥374.1 and ≥1, respectively). Assays performances together with complement status were compared regarding major organ involvement at the inclusion and flare-up risk prediction after follow-up. SLE-ELISpot showed the best performance in identifying active patients. High SLE-ELISpot results were associated with haematological involvement and, after follow-up, with an increased hazard ratio for disease flare-up (3.4) and especially renal flare (6.5). Additionally, the combination of hypocomplementemia and high SLE-ELISpot results increased those risks up to 5.2 and 32.9, respectively. SLE-ELISpot offers complementary information to anti-dsDNA autoantibodies to evaluate the risk of a flare-up in the following year. In some cases, adding SLE-ELISpot to the current follow-up protocol for SLE patients can improve clinicians’ personalised care decisions.

## 1. Introduction

Systemic lupus erythematosus (SLE) is a chronic autoimmune disease characterised by recurrent flares affecting several organs. Dysregulation of innate and adaptive immune responses leads to increased production of inflammatory cytokines, autoantibodies as well as immune complex deposition, which finally may cause severe organ damage [1]. The diagnosis of SLE, based on several clinical and laboratory parameters, is difficult due to the high heterogeneity of SLE manifestations among patients [2]. It was expected that autoantibodies detectable in SLE patients might be responsible for the wide range of disease manifestations and could be used to predict disease subsets and prognosis [3]. However, only IgG anti-double-stranded DNA (anti-dsDNA) together with IgG anti-Sm autoantibodies are included in SLE classification criteria since 1982 as highly specific autoantibodies [2,4,5,6]. Furthermore, complement levels and anti-dsDNA autoantibodies titres, both included in the SLE disease activity index (SLEDAI-2K) score [7], have been associated with clinical activity in SLE patients [8]. Nevertheless, anti-dsDNA autoantibodies titres are not enough to reflect the clinical activity of the disease in all patients [8]. Additionally, dsDNA autoantibodies can be directed to a wide range of different DNA-like structures with different specificities and avidities [9]. This fact could limit the autoantibody detection capacity of some assays which are using a specific dsDNA source.

The anti-dsDNA autoantibodies are produced by long-lived plasma cells in the bone marrow but, as demonstrated in lupus-prone mice and also in humans, short-lived antibody-secreting cells (ASC) also generate anti-dsDNA autoantibodies [10,11]. Thereafter, both short-lived ASCs and long-lived plasma cells participate in the production of anti-dsDNA autoantibodies in SLE [10,12,13]. The cytokine imbalance in SLE patients could affect both short-lived and long-lived ASC differentiation and survival [14,15,16]. Furthermore, it is known that short-lived ASCs are sensitive to most immunosuppressive treatments, including those directed to CD20 such as rituximab, but long-lived plasma cells are not. The latter are directly responsible for the persistent antibody levels in treated patients contributing to chronicity and disease relapses [12,17,18].

Better activity and organ involvement biomarkers are still needed to predict SLE flares or to evaluate B-cell targeting treatments [8]. Published data suggests that quantification of anti-dsDNA peripheral blood ASCs may be a promising biomarker for disease activity, major organ involvement and flare risk in SLE patients [19].

The B-cell ELISpot was first described in 1983 [20]. Since then, several studies have shown that the ELISpot assay can be used to detect B cell antigen-specific responses to different autoantigens like platelet membrane glycoprotein IIb-IIIa (gpIIb-IIIa) [21], dsDNA [17,22], glutamic acid decarboxylase [23] and insulinoma antigen 2 [23]. Kuwana et al. described a good correlation between the ELISpot quantification of ASC against anti-gpIIb-IIIa from the spleen and peripheral blood, supporting the use of peripheral blood mononuclear cells (PBMCs) for the B-cell ELISpot assay [21]. Although the ELISpot T-cell assay based on the detection of IFN-ɣ production upon antigen recognition of *Mycobacterium tuberculosis*-specific antigens [24] and SARS-CoV-2 antigens [25] is widely used, the use of antigen specific B-cell ELISpot currently remains limited to research and is still a manual, tricky and time-consuming assay.

The aim of the present study is to assess whether the anti-dsDNA ASC ELISpot (SLE-ELISpot) assay could provide better characterization of SLE patients in relation to disease activity or occurrence of flare ups compared with the usual commercially available anti-dsDNA autoantibodies assays used in autoimmunity laboratories for SLE management.

## 2. Patients and Methods

### 2.1. Patients Selection and Data Collection

We recruited a cohort of 91 individuals from our centre (Hospital Clínic de Barcelona) between January 2018 and February 2019 including 52 consecutive SLE patients, 21 patients diagnosed with other AID but without SLE (disease controls) and 18 healthy controls. The inclusion criteria were to fulfil the 2019 European League Against Rheumatism/American College of Rheumatology Classification Criteria for SLE [2] and patients being or who had been under B-cell treatment in the previous year were not included. Of the 52 SLE patients, 21 (40.4%) of them had an additional autoimmune disease (AID) being those Sjögren syndrome (n = 7), Sjögren syndrome and anti-phospholipid syndrome (n = 4), anti-phospholipid syndrome (n = 4), dermatomyositis (n = 2), Sjögren syndrome and dermatomyositis (n = 1), rheumatoid arthritis (n = 1), hypocomplementemic urticarial vasculitis (n = 1), and systemic sclerosis with dermatomyositis (n = 1). As controls, besides the 18 healthy individuals, we included 21 disease controls patients diagnosed with other AID but without SLE, such as rheumatoid arthritis (n = 9), Sjögren syndrome (n = 3), systemic vasculitis (n = 3), systemic sclerosis (n = 3), sarcoidosis (n = 1), anti-phospholipid syndrome (n = 1) and Crohn’s disease (n = 1).

Clinical features, demographic characteristics and laboratory parameters were collected from all patients. All SLE patients were followed by the same physicians (RC, GE, GMLL, SPG) at regular outpatient visits during the study period. Appointments occurred at least every 6 to 12 months or more frequently, depending on the clinical severity of each patient’s disease. At each appointment, symptoms and signs of SLE activity were checked together with laboratory abnormalities including blood cell count, renal and hepatic functions, urine profile with proteinuria and protein/creatinine ratio, levels of C3, C4 and CH50 and anti-dsDNA autoantibodies (these laboratory results are not analysed in this study). Specifically, those patients with active lupus nephritis were followed at the multidisciplinary unit by GE, MX and LQ. In this case, visits occurred every month during the first 3 to 6 months.

Active SLE disease was established by a clinical SLEDAI-2K > 4 at the moment of inclusion [7]. SLE flare was defined, based on an international consensus [26], as a measurable increase in disease activity (clinical SLEDAI-2K > 4 at the flare moment) involving new or worsening clinical signs and symptoms and/or laboratory measurements evaluated by the attending physician, leading to the consideration to change or increase treatment. Clinical SLEDAI-2K is an adaptation of SLEDAI-2K excluding the points coming from anti-dsDNA and complement status. Renal involvement was confirmed by biopsy-proven lupus nephritis according to the 2003 International Society of Nephrology/Renal Pathology Society (ISN/RPS) classification [27,28]. In addition, renal activity and chronic damage were determined using the National Institute of Health (NIH) activity and chronicity indexes, respectively [29].

### 2.2. Blood Samples

For the SLE-ELISpot, heparinised blood samples of SLE patients and controls were obtained at the time of inclusion. Within 6 h after blood extraction, PBMCs were isolated by Ficoll density gradient (Ficoll-Lymphoprep). Subsequently, PBMCs were frozen in 200 μL media and 200 μL DMSO solution (media: RPMI with 10% FBS and 10% Penicillin/Streptavidin; DMSO solution: PBS with 20% DMSO) and stored at −80 °C, until SLE-ELISpot was performed.

For anti-dsDNA autoantibodies determination, serum samples were obtained and stored in 1.5 mL Eppendorf tubes at −20 °C. Complement tests were performed with fresh sera obtained simultaneously with blood samples.

### 2.3. Anti-dsDNA Autoantibodies Detection and Complement Tests

IgG anti-dsDNA autoantibodies were quantified in serum samples by the commercial chemiluminescence immunoassay (CIA) (QUANTA Flash^®^; Inova Diagnostics, San Diego, CA, USA) in the BIO-FLASH^®^ instrument (Biokit, Barcelona, Spain). The manufacturer’s recommended cut-off was used (20 IU/mL). Moreover, IgG anti-dsDNA autoantibodies were analysed by *Crithidia luciliae* indirect immunofluorescence test (CLIFT) (Inova Diagnostics, San Diego, CA, USA), with sera diluted 1:10. Results were reported as negative or positive (positive results were arbitrary ranged from 1 to 8 according to fluorescence intensity at fluorescence microscopy in comparison with a positive control (1:160) by trained personnel). The reference value is negative.

Complement (C3, C4) serum levels and activity (CH50) were measured by turbidimetric immunoassay (Atellica CH C3 and Atellica CH C4; Siemens, New York, NY, USA and Autokit CH50; Fujifilm Wako Chemicals, Neuss, Germany, all are tested in Atellica CH Solution, Siemens, New York, NY, USA). Reference values were: 0.870–1.700 g/L for C3, 0.110–0.540 g/L for C4 and 28–60 U/mL for CH50. Hypocomplementemia was considered when either CH50 activity or C3 or C4 levels were lower than reference values. 

### 2.4. SLE-ELISpot Assay Conditions Standardization

From Hanaoka’s stated conditions, some other conditions were tested, and we selected the best ones (Appendix A). As a major change, the dsDNA was not treated with nuclease because no significant differences were observed with and without treatment. Additionally, the dsDNA coating strategy was changed from albumin coating to DNA Coating Solution^®^ (Thermo Fisher Scientific, Waltham, MA, USA). Afterwards, all samples collected at the time of inclusion were tested with the same conditions.

### 2.5. SLE-ELISpot Assay

Following the ELISpot design from Hanaoka [19] and the previous conditions tested, the PVDF plate (Millipore, Bedford, MA, USA) was sensitised with 100 μL calf thymus DNA (Thermo Fisher Scientific, Waltham, MA, USA) diluted with DNA Coating Solution (Thermo Fisher Scientific, Waltham, MA, USA) (100 μg/mL) following the distribution of the wells shown in Figure 1. After overnight incubation at 4 °C, the plate was washed with washing solution (PBS with 0.5 mM CaCl_2_ and 0.05% Tween 20) and blocked with the blocking solution (PBS with 5% FBS and 3% BSA) for 1 h 30 min. Patients’ PBMCs were thawed with RPMI media at 37 °C and viability was checked in a Neubauer chamber with Trypan Blue solution. Patients’ PBMCs (50 μL/well at 10 × 10^6^ PBMCs/mL), as well as undiluted sera samples (from patients and controls), were incubated for 4 h at 37 °C. Afterwards, plates were washed with washing solution and incubated with alkaline phosphatase-conjugated goat anti-human IgG antibody (Thermo Fisher Scientific, Waltham, MA, USA) (80 μL/well, diluted 1:1000 in PBS-Ca) for 2 h. Following incubation with Nitro Blue Tetrazolium solution (Thermo Fisher Scientific, Waltham, MA, USA) (100 µL/well) for 7–10 min, anti-dsDNA IgG immune-complexes identified by the anti-human IgG were visualised as spots. After stopping the reaction using tap water, the plate was dried in a dry oven for at least 1 h at 37 °C and stored thereafter at 4 °C. Spots were counted by an automatic EliSpot Reader (AID EliSpot ELRIFL04) using the EliSpot Reader version 7.0 (AID GmbH, Penzberg, Germany) software.

In order to assure proper dsDNA plate sensitisation and correct specificity, we introduced the following modifications: (1) we tested PBMCs and sera controls in non-dsDNA-sensitised wells in addition to dsDNA-sensitised wells and (2) we included one negative control (culture media) and three positive: one positive serum for anti-dsDNA autoantibodies by CIA and CLIFT and two single positive sera either by CIA or CLIFT.

### 2.6. Statistics

The IBM SPSS Statistics PC package (Version 22) was used for statistical analyses. Results for *p* values < 0.05 were considered statistically significant. Cut-off values were established through the receiver operating characteristics (ROC) curve analysis and the highest Youden Index value was chosen. The continuous data were described by the mean and standard deviation (SD) or median and 25th75th interquartile range (IQR), according to its distribution. Categorical variables were expressed as absolute numbers and percentages. Normality was assessed by the Shapiro–Wilk test or Kolmogorov–Smirnov test depending on the sample size. Differences in proportions were analysed using the χ² test or Fisher’s exact test when the expected counts were ≤5. Differences in means of continuous variables were analysed using the parametric Student t-test or the Mann–Whitney nonparametric U test when the variable distribution was not normal. Correlations were calculated with Pearson’s R, Spearman’s coefficient or Kendall’s tau b depending on the type of variable. The odds ratio (OR) and 95% confidence interval (95%CI) were also calculated. We analysed the presence of a SLE flare for 12 months follow-up by the Kaplan–Meier curves for each condition and potential significant differences were evaluated by the log-rank test. Finally, hazard ratio (HR) and 95%CI were calculated.

## 3. Results

### 3.1. Study Cohort

The study cohort consisted of 91 individuals. There were no significant differences in age or sex between SLE patients (SLE [n = 31] and overlap SLE-AID [n = 21]) and controls (healthy [n = 18] and AID individuals [n = 21]). Clinical features, including demographic characteristics, disease activity measured by clinical SLEDAI-2K, major organ involvement (including renal, articular, haematological, cutaneous involvement and serositis) and disease flares as well as laboratory parameters are shown in Table 1. No differences were found either in clinical SLEDAI-2K, active disease, major organ involvement, anti-dsDNA autoantibodies (measured by CIA and CLIFT) or hypocomplementemia between SLE and overlap SLE-AID groups. Therefore, all patients with SLE, alone or overlapping with another AID, have been analysed together in a single group.

### 3.2. SLE-ELISpot Improved Assay

Based on the modifications made to the ELISpot described by Hanaoka et al. [19], the design of our SLE-ELISpot assay is shown in Figure 1. Results, ranging from 0 to ∞, were calculated according to the following formula attending to PBMCs wells: $$\mathrm{SLE}\text{-}\mathrm{ELISpot}\;\mathrm{result}={\bar{x}}_{sens}-{\bar{x}}_{non\text{-}sens}$$
$\bar{x}$*_sens_*: average number of spots in dsDNA-sensitised wells$\bar{x}$*_non-sens_*: average number of spots in non-dsDNA-sensitised wells


Wells with sera samples were also included in the plate as controls of the technical procedure.

### 3.3. Clinical Performance of SLE-ELISpot, CIA and CLIFT

Two different reference values were established for all three assays through the ROC curves analysis choosing the highest Youden Index value: (1) a “diagnostic cut-off” comparing SLE patients with healthy controls and (2) an “activity cut-off” comparing active and non-active SLE patients.

In our cohort, the diagnostic cut-off for CIA and CLIFT coincided with the manufacturer recommended cut-off (≥20 IU/mL for CIA and ≥1 (negative or positive) for CLIFT). The sensitivity and specificity were 87.0% and 88.9% for CIA and 64.8% and 100% for CLIFT. We established the SLE-ELISpot diagnostic cut-off in ≥1.8 spots with 64.8% sensitivity and 77.8% specificity.

As expected, CIA and CLIFT showed much better sensitivity and specificity for SLE diagnosis than SLE-ELISpot (Figure 2A). Therefore, SLE-ELISpot was not able to improve the current diagnostic tools.

Regarding the activity cut-off, established through the comparison of active and non-active patients, SLE-ELISpot activity cut-off (≥11.24 spots) had a sensitivity and specificity of 57.1% and 83.9%, respectively. Conversely, CIA (≥374.1 IU/mL) and CLIFT (≥1) activity cut-offs showed 42.9% and 71.4% sensitivity and 96.8% and 41.9% specificity, respectively (Figure 2A). Values for CIA, CLIFT and SLE-ELISpot of active and non-active patients, are shown in Figure 2B.

Thus, SLE-ELISpot activity cut-off showed a better Youden Index than CIA and CLIFT to identify active SLE patients (0.410 vs. 0.397 and 0.133, respectively). 

Consequently, for the rest of this paper, we focused on the activity cut-off. A result above the activity cut-off will be referred to as “high” and a result under the activity cut-off will be referred to as “low”.

### 3.4. Association with Major Organ Involvement in SLE Patients

At the time of inclusion in the study, lupus nephritis (n = 28) was the most common organ involvement followed by joint (n = 8), haematological (n = 8) and skin involvement (n = 7), and the presence of serositis (n = 7) (Table 1). Hypocomplementemia was associated with lupus nephritis (OR 7.3 [95%CI 2.1–25.3]). A high CIA result was associated with articular (OR 6.3 [95% CI 1.2–32.4]) and haematological involvement (OR 6.3 [95%CI 1.2–32.4]). A high SLE-ELISpot result was associated with haematological involvement (OR 23.8 [95%CI 2.6–217.1]). Additionally, SLE-ELISpot but not CIA results were higher in patients with haematological involvement than in patients without (median 16.73 (13.5–20.2) and 2.60 (0.0–10.7); *p* < 0.001, respectively). Finally, a high SLE-ELISpot result combined with hypocomplementemia was associated with articular involvement with an OR = 6.3 (95%CI 1.4–26.9) (Figure 3). No other combination was associated with major organ involvement.

### 3.5. Flare Prediction Capacity

Regarding the 52 SLE patients, 14 (26.9%) presented with a flare up during the 12-month follow-up. Eight (57.1%) of these patients had a high SLE-ELISpot result while six (42.9%) had a low SLE-ELISpot result. Moreover, 29 out of 35 (82.9%) low SLE-ELISpot patients did not have a flare up while eight out of 17 (47.1%) high SLE-ELISpot patients presented with a flare up in the following year. Thus, patients with a high SLE-ELISpot result had a significantly increased hazard ratio (HR) to present with a flare up (HR: 3.4 [95%CI 1.1–10.5]) (Figure 4). The combination of hypocomplementemia and a high SLE-ELISpot was related to a high risk of a flare up (HR: 5.2 (95%CI 1.3–20.4)) (Figure 4). Neither high CIA, CLIFT or hypocomplementemia alone were significantly associated with an SLE flare up in a 1-year follow-up (Figure 4).

Among the 14 SLE patients with a flare up, seven (50%) presented biopsy-proven kidney involvement. Two of them had a low SLE-ELISpot while five had a high result. Hypocomplementemia at baseline was present in all seven patients with lupus nephritis at the time of the flare up. A high SLE-ELISpot result implies an increased HR for patients to have a kidney involved flare up (HR: 6.5 (95%CI 1.3–32.2)) (Figure 5). Additionally, the combination of hypocomplementemia and a high SLE-ELISpot raises the HR to 32.9 (95%CI 4.9–221.3) (Figure 5). Conversely, high CIA or CLIFT values were not significantly associated with an increased risk of a renal flare up at the 1-year follow-up (Figure 5).

Furthermore, nine patients presented with a flare up involving joints. However, none of the tested methods, alone or in combination, were predictive of an articular flare up (Figure 6). Cutaneous (n = 2) and haematological (n = 2) flare ups were not analysed due to low incidence.

## 4. Discussion

This study aimed to analyse whether the quantification of anti-dsDNA autoantibodies secreting cells (ASC), detected by SLE-ELISpot in blood samples of patients with SLE, could provide any advantage for disease activity or prognostic evaluation, short-term flare up prediction or major organ involvement.

It is known that anti-dsDNA autoantibodies are produced by short-lived ASCs localised in peripheral blood, in addition to long-lived plasma cells, mainly absent in peripheral blood [10]. A good correlation between peripheral blood and spleen antigen-specific ASCs quantification has been shown [21]. Thus, the quantification of ASCs in peripheral blood could add relevant clinical information regarding disease activity or flare up prediction. 

In the first part of the study, we introduced some modifications to the ELISpot assay designed by Hanaoka [19]. Firstly, to achieve a greater specificity of the assay, non-dsDNA-sensitised wells were included in the SLE-ELISpot design allowing the SLE-ELISpot result to be calculated as the subtraction of the non-dsDNA-sensitised wells’ number of spots from the spots obtained on the dsDNA-sensitised wells. Moreover, two different cut-off values were established to assess the clinical value of the quantification of specific anti-dsDNA ASC from peripheral blood: one directed to diagnosis accuracy, named “diagnosis cut-off”, and a second one to better identify disease activity named “activity cut-off”. As expected, using the diagnostic cut-off (≥1.8 spots), SLE-ELISpot was far less sensitive for diagnosis than IgG anti-dsDNA autoantibodies determination by CIA and CLIFT. However, in the present study SLE-ELISpot was the best method to identify active patients. Although it needs to be confirmed in larger cohorts, these results suggest that the SLE-ELISpot assay may help to better stratify active and non-active patients.

In the second part of the study, we studied the association between SLE-ELISpot, CIA and CLIFT results with clinical manifestations at inclusion time and the flare up prediction capacity in a 1 year follow-up. 

In SLE patients, lupus nephritis constitutes one of the most severe organ manifestations that can lead to end-stage renal disease and even require renal replacement therapy. Indeed, renal involvement is the main predictor of mortality in SLE patients [30]. According to the literature, IgG anti-dsDNA autoantibodies titres and low complement C3/C4 are associated with lupus nephritis and recurrent disease flares [31,32]. However, it is also well known that IgG anti-dsDNA autoantibodies at high titres can be also observed in patients in remission [33]. In our study, the previously known correlation of hypocomplementemia with the presence of lupus nephritis (OR: 7.3) was confirmed. In relation to IgG anti-dsDNA autoantibodies, detected by CIA, high results were associated with articular and haematological involvement (OR: 6.3 for both). Regarding other major organ involvement, high SLE-ELISpot results were strongly associated with haematological disturbances (OR: 23.8), and the combination of high SLE-ELISpot results and hypocomplementemia was associated with articular involvement (OR: 6.3). The differences in these associations could be due to different pathogenic mechanisms but this fact must be confirmed in other studies including a higher number of patients.

During the follow-up of SLE patients, tools for flare risk prediction constitute a need not yet covered, which may facilitate better disease monitoring as well as an earlier treatment of patients. Hanaoka previously suggested that anti-dsDNA ELISpot may help to predict the risk of SLE flare ups in the following year [19]. The results of our study corroborate this suggestion, as a high SLE-ELISpot result showed an HR of 3.4 for SLE patients to have a flare up (*p* = 0.043). Beyond the prediction of flare risk, our results indicate that this prediction is significantly higher for flare ups with renal involvement (HR: 6.5). Furthermore, the addition of hypocomplementemia to SLE-ELISpot brings an even better prediction capacity, increasing the HR for any kind of flare up and renal flare up to 5.2 and 32.9, respectively. As noted, SLE-ELISpot was better than CIA and CLIFT to predict a flare up at 1-year and even better for renal flare ups. Moreover, whereas the SLE-ELISpot positive predictive value concerning a flare was good (47.1% of patients with high results had a flare), the negative predictive value for ELISpot activity cut-off was even better (82.9% of low patients did not present a flare). Thus, the addition of SLE-ELISpot to the anti-dsDNA autoantibody determination in some complex SLE patients may be of interest.

Nevertheless, our study has some limitations. First, patients were not collected at SLE onset, most of them were under immunosuppressive treatments and we could not assess the impact of this condition on flare appearance. Second, the number of consecutively included was low. Our hospital is a reference centre in the diagnosis and treatment of patients with autoimmune diseases. This may be the reason for the high percentage of patients with overlapping syndromes due to the diagnostic and therapeutic complexity they entail. On the other hand, this overlap is not uncommon and is described in up to 20% of patients with SLE in the Registry of the Spanish Society of Rheumatology (RELESER) [34]. We think that the type of patients included is a picture of daily clinical practice in real life. Third, we chose the SLEDAI-2K score to evaluate disease activity and is known that it has some limitations, for example, it overemphasises some symptoms whereas others are absent. Moreover, SLEDAI-2K only scores the presence or absence of each item without a severity spectrum evaluation of them [35]. Fourth, the SLE-ELISpot assay is manual, tricky and time-consuming and requires properly trained personnel. For this reason, it cannot be used as an automatic daily-routine assay at a large scale. However, our study shows that for some specific patients who require a closer follow-up, SLE-ELISpot can provide valuable information regarding their clinical activity and may help physicians to make therapeutic decisions. For this reason, the fact that this work is based on consecutive clinical and laboratory monitoring SLE patients’ samples is a remarkable strength. Another strength is that this study is made with the baseline ELISpot results, this allows the ASC study to be feasible. However, periodic and systematic ELISpot studies to know the evolution of specific B-cells may be of interest and it should be addressed. Besides, we designed this study with the objective to mimic physiological conditions, for this reason B cells were not pre-activated and the number of cells in the well was not normalised according to the B cell proportion of each patient. Although these results seem promising, they are derived from post-hoc analyses with a relative low number of patients and must be confirmed in larger cohorts.

Additionally, in recent years, some authors had studied the detection of antigen-specific B-cells by flow cytometry [36,37,38]. Since flow cytometry techniques are available in most immunology laboratories, it might be of interest to compare the presence of anti-dsDNA secreting cells detected by SLE-ELISpot with dsDNA-specific B-cells detected by flow cytometry. Moreover, B cell are somehow regulated by T cells, thus the study of both B and T-specific cells could be also of interest. In fact, in this study the use of PBMCs assure that all subpopulations are included.

In conclusion, according to our results, SLE-ELISpot assay may offer relevant and complementary information to that obtained by IgG anti-dsDNA autoantibodies quantification, particularly related to the risk of a new disease flare up in the 1-year follow-up but is not suitable to replace the IgG anti-dsDNA autoantibodies determination for SLE diagnosis. Indeed, this study suggests that the addition of the SLE-ELISpot assay to the current follow-up protocol may improve options for early using targeted therapies to reduce chronic organ damage in some patients despite standard of care therapy, based on the association of SLE-ELISpot over activity cut-off results with the presence of active disease or the risk of flare up, especially in the case of renal flare ups. Finally, larger prospective studies are needed to further confirm the association with clinical activity and the flare up predictability of the SLE-ELISpot assay.

## Figures and Tables

**Figure 1 jcm-12-01295-f001:**
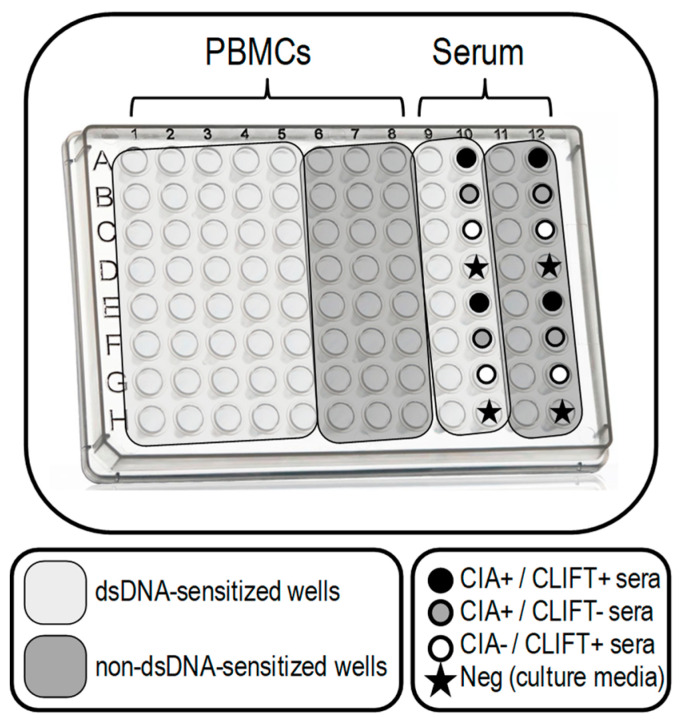
SLE-ELISpot plate design. SLE-ELISpot assay was performed with PBMCs (columns 1 to 8). Sera samples (columns 9 to 12) were included as a control for the technical procedure (being columns 9 and 11 for patients sera and columns 8 and 10 for selected controls with a known result for CIA and CLIFT). Wells in light grey are dsDNA-sensitised. Dark grey wells are non-dsDNA-sensitised. Each row was used for one patient.

**Figure 2 jcm-12-01295-f002:**
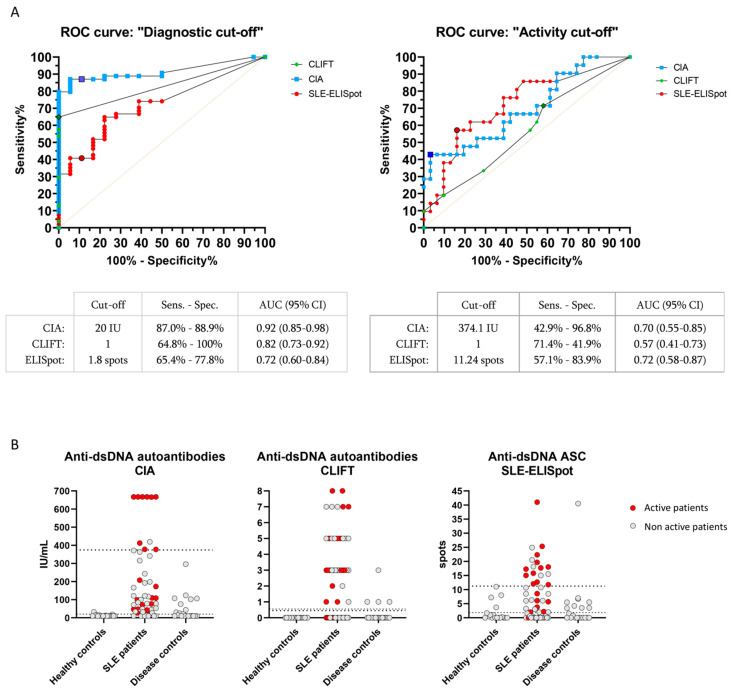
CIA, CLIFT and SLE-ELISpot results: ROC curves and values distribution. ROC curves analysis using diagnostic cut-off (**A**, left) and activity cut-off (**A**, right) for CIA, CLIFT and SLE-ELISpot. Bordered signs indicate selected cut-off points. Values distribution for CIA (**B**, left), CLIFT (**B**, middle) and SLE-ELISpot (**B**, right). Dashed lines: diagnostic cut-off. Pointed lines: activity cut-off. Red signs indicate active SLE patients. Grey signs indicate non-active SLE patients. Abbreviations: AUC (Area under the curve); Sens (Sensitivity); Spec (Specificity).

**Figure 3 jcm-12-01295-f003:**
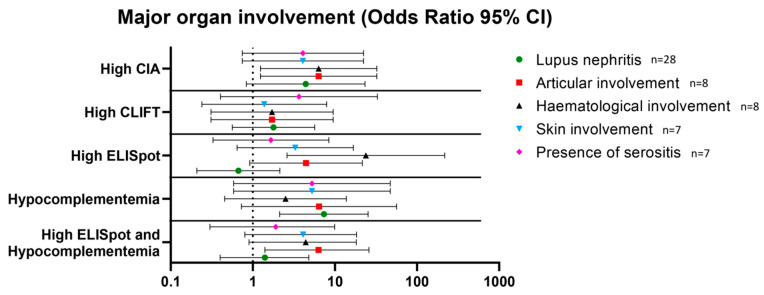
Association between high levels of anti-dsDNA antibodies measured by CIA, CLIFT and ELISpot and hypocomplementemia and major organ involvement. Results are expressed as odds ratios and 95% confidence intervals. Abbreviations: CIA: chemiluminescence immunoassay; CLIFT: *Crithidia luciliae* indirect immunofluorescence test.

**Figure 4 jcm-12-01295-f004:**
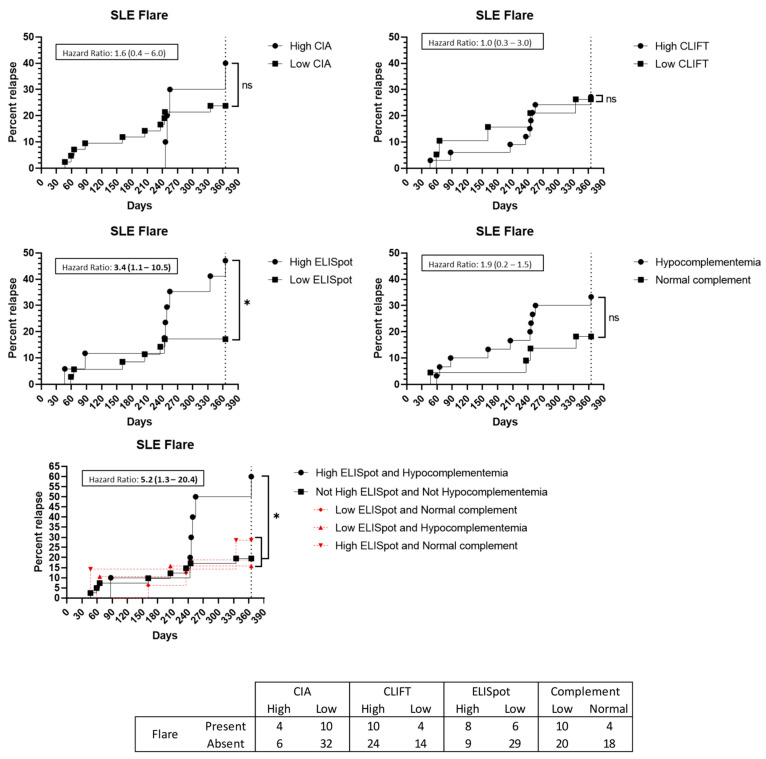
Flare up prediction capacity: Kaplan–Meier survival analysis and hazard ratio. X-axis: days from sampling to flare up. Y-axis: percentage of patients presenting with a flare up. ns: non-significant. *: *p* < 0.05. Abbreviations: CIA: chemiluminescence immunoassay; CLIFT: *Crithidia luciliae* indirect immunofluorescence test; SLE: systemic lupus erythematosus.

**Figure 5 jcm-12-01295-f005:**
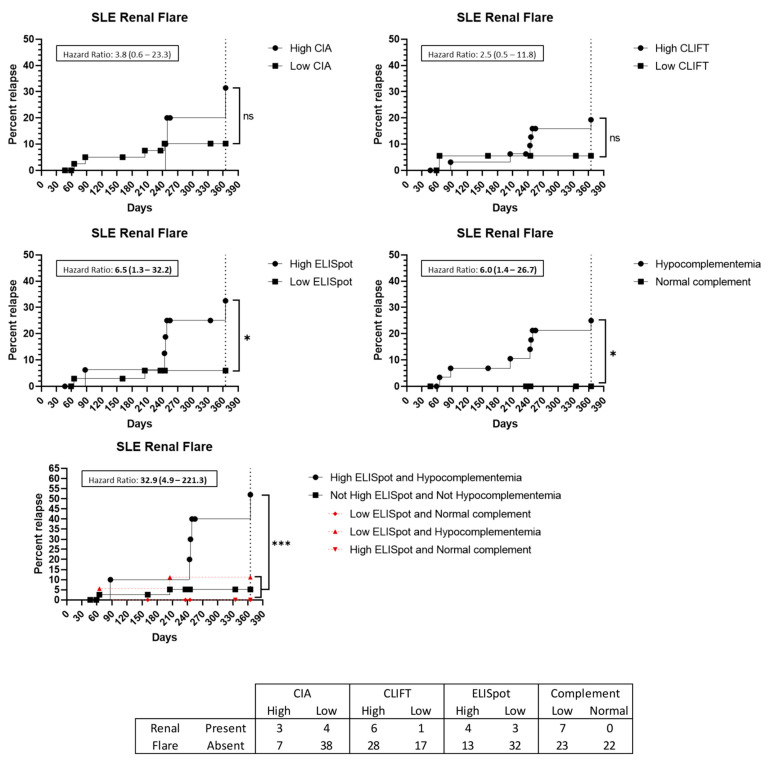
Renal flare up prediction capacity: Kaplan–Meier survival analysis and hazard ratio. X-axis: days from sampling to flare up. Y-axis: percentage of patients presenting with a flare up. ns: non-significant. *: *p* < 0.05. ***: *p* < 0.001. Abbreviations: CIA: chemiluminescence immunoassay; CLIFT: *Crithidia luciliae* indirect immunofluorescence test; SLE: systemic lupus erythematosus.

**Figure 6 jcm-12-01295-f006:**
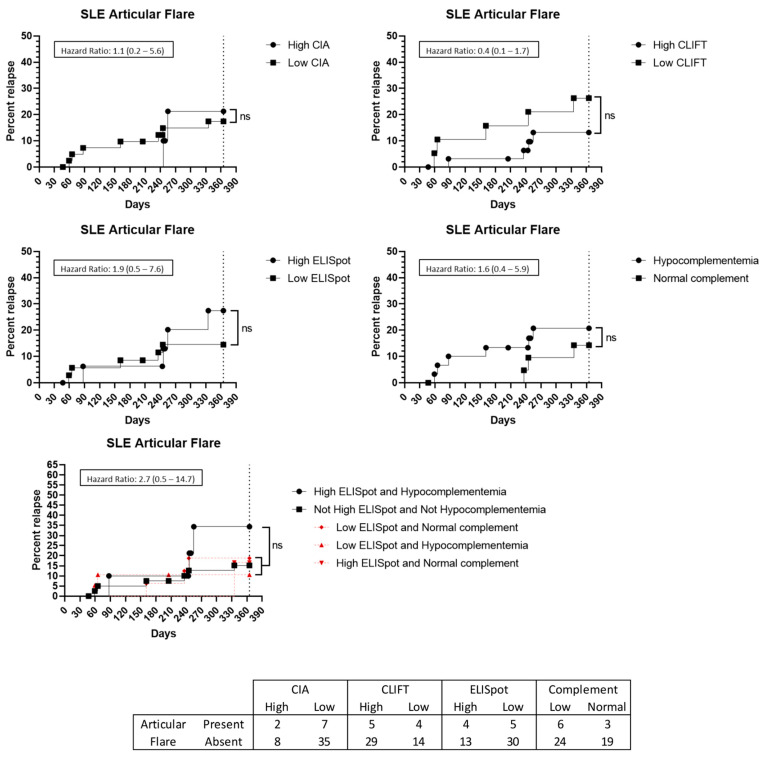
Articular flare up prediction capacity: Kaplan–Meier survival analysis and hazard ratio. X-axis: days from sampling to flare up. Y-axis: percentage of patients presenting with a flare up. ns: non-significant. Abbreviations: CIA: chemiluminescence immunoassay; CLIFT: *Crithidia luciliae* indirect immunofluorescence test; SLE: systemic lupus erythematosus.

**Table 1 jcm-12-01295-t001:** Demographic and clinical characteristics of the study cohort.

	Control Group		SLE Group		Intra-SLE Group (*p* Value)
	Healthy Controls (n = 18)	AID Controls (n = 21)	SLE (n = 31)	SLE Overlapping Other AID (n = 21)	
Sex: female, n (%)	16 (88.9%)	17 (81.0%)	29 (93.5%)	19 (90.5%)	0.999
Age, mean ± SD	32 ± 10	55 ± 13	41 ± 15	46 ± 16	0.183
Anti-dsDNA autoantibodies (CIA), mean ± SD	12.4 ± 5.6	49.6 ± 68.9	209.0 ± 223.4	163.8 ± 192.8	0.441
Anti-dsDNA autoantibodies (CLIFT), n (%)	0 (0.0%)	4 (19.0%)	21 (67.7%)	13 (61.9%)	0.769
Hypocomplementemia, n (%)	-	-	17 (54.8%)	13 (61.9%)	0.776
clinical SLEDAI-2K, median [IQR]	-	-	4 [2–13]	4 [4–8]	0.784
Active disease (clinical SLEDAI-2K > 4), n (%)	-	-	13 (41.9%)	8 (38.1%)	0.999
Major organ involvement (at sampling)					
Renal involvement, n (%)	-	-	19 (61.3%)	9 (42.9%)	0.259
Active, n (%)	-	-	10 (45.5%)	6 (33.3%)	0.526
Chronic, n (%)	-	-	16 (57.1%)	8 (40.0%)	0.380
Articular involvement, n (%)	-	-	4 (12.9%)	4 (19.0%)	0.700
Haematological involvement, n (%)	-	-	5 (16.1%)	3 (14.3%)	0.999
Cutaneous involvement, n (%)	-	-	4 (12.9%)	3 (14.3%)	0.999
Presence of serositis, n (%)	-	-	5 (16.1%)	2 (9.5%)	0.687
Patients with a flare at follow-up, n (%)	-	-	8 (25.8%)	6 (28.6%)	0.999
Patients with renal flare, n (%)	-	-	4 (12.9%)	3 (14.3%)	0.999
Days between sample and flare, mean ± SD	-	-	238 ± 99	150 ± 88	0.285

Abbreviations: AID: autoimmune diseases; SLE: systemic lupus erythematosus; clinical SLEDAI: SLE disease activity index.

## Data Availability

The raw data supporting the conclusions of this article will be made available by the authors, without undue reservation.

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
