# Peer review of "Anti-dsDNA B-Cell ELISpot as a Monitoring and Flare Prediction Tool in SLE Patients"

_jcm, 2023, doi:10.3390/jcm12041295_

Round 1

Reviewer 1 Report

See enclosed file

Reviewer 2 Report

In the submitted manuscript the authors analysed the diagnostic value of an ELISpot assay aiming at detecting anti-dsDNA autoantibody producing cells in PBMCs from SLE patients. According to the presented data the assay may have some diagnostic advantages and might help to identify patients at risk for renal flares. Overall, this is surely interesting data, but I have several concerns to be addressed:

1.    Embedding of the figures was not satisfying (might be a problem of the submission process) as the size and location does not fit well.

2.    The introduction/methods read as if antibody assays were performed at several time points per patient. Looking at the results, it looks as if there was one initial time point for antibody determination followed by primarily clinical follow-up visits of up to one year. This part needs clarification, maybe ad a table/flow-chart describing blood sampling and visits (how many patients/time point were followed, indicate when sampling for ELISpot etc. ?)

3.    With regard to the cross-sectional part 3.6 (association with organ involvement), the combination of high ELISpot with hypocomplementaemia is not better than using high CIA as a diagnostic parameter. Furthermore, combination of ELISpot with hypocomplementaemia changes the diagnostic sensitivity from hematological involvement to articular involvement, which not necessarily means that the combination strategy is improving the test performance. It rather leads to the identification of other patients/manifestations. In addition, both diagnostic hints do not fit with high ELISpot+hypocomplementaemia being indicative of later renal flare (but neither articular nor hematological). Overall, these discrepancies might point to either weak diagnostic performance (leading to a high variability within the patient cohort) or to different pathogenic mechanisms being involved in hematological, articular and renal involvement respectively. This issue needs to be better discussed and added to the list of limitations requiring confirmatory studies.

4.    Paragraph 3.7: Last two sentences of the first paragraph and third paragraph seem to be redundant.

5.    Figure 5 shows graphs for articular flares (not renal as written) and these graphs are identical to the ones in Figure 6

Reviewer 3 Report

The manuscript by Perez-Isidro, et al., seeks to determine if use of anti-dsDNA ELISpot assays would be predictive of flares or for monitoring disease in SLE. The investigators compared healthy controls with SLE patients, and patients with other autoimmune diseases. A subgroup of SLE patients with other autoimmune complications was also assessed. The statistics and design of the experiments appears solid and may be of interest to other investigators. A few criticisms might be made and are listed below:

1)      Although there is some discussion at the end of the article regarding the likelihood that other less cumbersome assays than ELISpot (including flow cytometry) may eventually be more clinically useful, there is little discussion about implications of the findings or about potential caveats of the study.

2)      It is interesting that the time to flare averages more than 240 days from the time cells expressing anti-DNA antibodies are detected. This seems a bit long to be directly associated with the assessment of antibody-producing cells.

3)      A potential complication of these results is that patients with more dsDNA-producing B cells by ELISPot may also simply have increased proportions of total Ig-producing B cells and that the flares are not directly linked to dsDNA production in this study since total PBMCs are used rather than total B cells. Indeed, the contribution of accessory cells such as T cells that regulate those B cells may have more direct influence on the data. Thus, it seems premature to directly link these data only to dsDNA-producing B cells.

Round 2

Reviewer 1 Report

The paper has not improved much as the main issues remain regarding 

1) an odd selection of patients many of which have overlap/other syndromes where anti-dsDNA Ab presence likely confounded classification.

2) While it was obvious the Elispot had no diagnostic value, the authors keep insisting it has merit in disease monitoring based on their flawed circular reasoning by applying posthoc analyses to predict flares.

3) defining active SLE based on clinical SLEDAI (although this is not clearly mentioned) , while a flare was based on clinical judgement. As the outcome is different from starting point, this makes it even harder to interpret the already confusing data. The authors throughout the paper mix the term active disease (which only applies to baseline data) with flare  (which happened 6- 9 months after baseline and had a different definition). 

4) As expected none of the anti-dsDNA test reached a good (kappa based) agreement with SLEDAI >4 suggesting none of the antidsDNA Ab assays was clinically relevant and undermining the authors conclusion.

5) The finding that hypocomplementemia as a measure of pathogenic (ie nephritogenic) autoantibodies incl anti-dsDNA Ab is possibly the main finding of this study , although already well known ans studied.  

Author Response

Please see the attachment. We also upload a Word version with tracking changes and a PDF version with all changes accepted and the figures and tables well disposed.

Reviewer 2 Report

I could not find a corrected version with marked changes (between the first and the second version) making it very difficult to judge how the authors adapted the manuscript. 

I still think that the differences in test performance with regard to clinical manifestation at the time of sampling versus clinical manifestation at follow-up can be due to the small sample size of the study limiting the accuracy of the analyses – and not necessarily due to different pathogenic mechanisms. This option should be added to the list of limitations (or be part of the discussion)

Author Response

(The authors gave the same response as above.)
